# Sticky collisions of ultracold RbCs molecules

Philip D. Gregory[1], Matthew D. Frye [2], Jacob A. Blackmore [1], Elizabeth M. Bridge [1], Rahul Sawant[1], Jeremy M. Hutson [2] & Simon L. Cornish [1]

Understanding and controlling collisions is crucial to the burgeoning field of ultracold molecules. All experiments so far have observed fast loss of molecules from the trap. However, the dominant mechanism for collisional loss is not well understood when there are no allowed 2-body loss processes. Here we experimentally investigate collisional losses of nonreactive ultracold $^{87}Rb^{133}Cs$ molecules, and compare our findings with the sticky collision hypothesis that pairs of molecules form long-lived collision complexes. We demonstrate that loss of molecules occupying their rotational and hyperfine ground state is best described by second-order rate equations, consistent with the expectation for complex-mediated collisions, but that the rate is lower than the limit of universal loss. The loss is insensitive to magnetic field but increases for excited rotational states. We demonstrate that dipolar effects lead to significantly faster loss for an incoherent mixture of rotational states.

[1] Joint Quantum Centre (JQC), Durham—Newcastle, Department of Physics, Durham University, Durham DH1 3LE, UK. [2] Joint Quantum Centre (JQC), Durham—Newcastle, Department of Chemistry, Durham University, Durham DH1 3LE, UK. Correspondence and requests for materials should be addressed to J.M.H. (email: j.m.hutson@durham.ac.uk) or to S.L.C. (email: s.l.cornish@durham.ac.uk)

A growing number of experiments now produce ground-state polar molecules at ultracold temperatures, either by associating pairs of atoms[1–8] or by direct laser-cooling of molecules[9,10]. These experiments offer an exciting new platform for the study of ultracold dipolar gases[11–15] and quantum-state-controlled chemistry[16–19]. For molecules produced by association, the sample densities are sufficiently high that molecular collisions are important and measurable. Yet a proper understanding of ultracold molecular collisions remains elusive.

For ultracold atomic systems, a detailed understanding of collisions has been developed through decades of research comparing theory and experiment[20,21]. In particular, the control of interactions through intra-species magnetic Feshbach resonances[21–24], with quantitative calculations of the scattering length, has proved crucial. For example, it has allowed study of the BEC–BCS crossover in Fermi gases[25–31] and of Efimov physics[32–42]. A detailed understanding of collisions will be equally crucial in future experiments with ultracold molecular gases.

There has been relatively little comparison of experiment and theory for molecular collisions. Many alkali dimers can undergo exoergic two-body exchange reactions of one or more of the types

$$
\begin{aligned}
2XY &\rightarrow X_2 + Y_2; \\
2XY &\rightarrow X_2Y + Y; \\
2XY &\rightarrow X + XY_2.
\end{aligned}
\tag{1}
$$

Ultracold collisions between such molecules have been studied experimentally in KRb[15,43,44], NaLi[7] and triplet Rb$_2$[45]. Collisional loss was found to occur with high probability for molecular pairs that reach short range, and was attributed to the reactions (1). However, there are other alkali dimers, such as NaRb and RbCs in their vibronic ground states, for which all the reactions (1) are energetically forbidden[46]. Surprisingly, these also show high collisional loss rates. For example, Ye et al.[47] compared the loss rate for NaRb molecules in the ground and first-excited vibrational states, and found high loss and heating rates regardless of the energetics of the reactions.

One possible mechanism for fast losses of chemically stable species has been proposed by Mayle et al.[48,49]. They argue that the large number of rovibrational states available supports a dense manifold of Feshbach resonances. Resonant collisions may form long-lived two-molecule collision complexes. A further collision between a complex and a molecule can then lead to loss of all three molecules from the trap. Complexes may also be lost by other mechanisms. Complex formation may produce second-order kinetics even if the loss is three-body in nature. Nevertheless, the three-body loss is effectively enhanced by the long lifetime of the complexes. We refer to this as the sticky collision hypothesis.

The lifetime $\tau$ of a collision complex is related to the resonance width $\Gamma$ by $\tau = \hbar/\Gamma$. The model of Mayle et al. assumes that the mean width is $\langle \Gamma \rangle = N_o/2\pi\rho$, where $\rho$ is the density of states and $N_o$ is the number of open channels for the free molecular pair. This is based on Rice–Ramsperger–Kassel–Marcus (RRKM) theory[50] and effectively assumes that the motion is ergodic, i.e., that energy is fully randomised in the complex. For collisions of RbCs in the rovibrational ground state, $N_o = 1$ and the predicted density of states is $\rho/k_B = 942\ \mu K^{-1}$ ($\rho/\mu_{mag} = 368\ G^{-1}$)[49]; this gives a sticking lifetime of 45 ms.

In the following, we test the model of Mayle et al. by measuring loss from an optically trapped sample of ground-state RbCs molecules. These molecules are chemically stable against all available two-body atom-exchange reactions[46], yet fast losses are still observed. We demonstrate that the loss is best described by a rate equation that is second-order in the density. We investigate the temperature dependence of the loss in the rotational and hyperfine

ground state, and compare our results to a single-channel model that uses an absorbing boundary condition to take account of short-range loss[51,52]. We find a significant difference between the measured loss rates and those expected in the universal limit, in which all two-body collisions that reach short range lead to loss. We then increase the internal energy of the molecule, both by varying the magnetic field and by preparing the molecules in excited rotational and/or hyperfine states, and observe similar loss rates. Finally, we prepare the molecules in an incoherent mixture of ground and first-excited rotational states. In this mixture we observe a much faster loss than for molecules in a single state. Taken together, our measurements support the sticky collision hypothesis, but with a rate lower than predicted by the full model of Mayle et al. This may arise from a breakdown of ergodicity, manifested as an average width smaller than predicted by RRKM theory and perhaps due to a geometrical restriction on complex formation.

## Results

**Measuring loss due to molecule–molecule collisions.** Our experiments are performed with a gas of $X^1\Sigma^+$ RbCs molecules, initially occupying the rovibrational and spin-stretched hyperfine ground state $|N = 0, M_N = 0, m_i^{Rb} = 3/2, m_i^{Cs} = 7/2\rangle$ at a magnetic field of 181.5 G. Here, $N$ is the rotational quantum number with projection $M_N$ along the quantisation axis, and $m_i^{Rb}$ and $m_i^{Cs}$ are the atomic nuclear spin projections. The molecules are confined to an optical dipole trap (ODT) with typical initial temperature 1.5(1) μK and peak density of $1.9(2) \times 10^{11}\ cm^{-3}$. We observe loss of molecules as a function of hold time in the ODT as shown in Fig. 1. A molecule is considered 'lost' either if it is ejected from the trap or if it is in a state other than that in which it was prepared (including a complex).

To characterise the dominant loss mechanism, we model the rate of change of density $n$ as $\dot{n}(\mathbf{r}, t) = -k_\gamma n(\mathbf{r}, t)^\gamma$, where the power of the density $\gamma = 1$, 2 or 3 corresponds to losses where the rate-determining step is a one-, two- or three-body process, respectively. We numerically solve the coupled rate equations

$$
\begin{aligned}
\dot{N}_{mol}(t) &= -k_\gamma C^{(\gamma-1)} \left( \frac{N_{mol}(t)^\gamma}{\gamma^{3/2} T(t)^{(3/2)(\gamma-1)}} \right), \\
\dot{T}(t) &= k_\gamma C^{(\gamma-1)} \left( \frac{\gamma - 1}{2\gamma} \right) \left( \frac{N_{mol}(t)^{\gamma-1}}{\gamma^{3/2} T(t)^{(3\gamma-5)/2}} \right),
\end{aligned}
\tag{2}
$$

and fit the variation in number with $\gamma$ and $k_\gamma$ as free parameters. Here, $N_{mol}(t)$ is the number of molecules remaining in the initial state, $T(t)$ is the temperature of the remaining distribution, and $C = (m\bar{\omega}^2/2\pi k_B)^{3/2}$, where $m$ is the mass of the molecule and $k_B$ is the Boltzmann constant. In deriving Eq. (2), it is assumed that the molecules remain in thermal equilibrium and that $k_\gamma$ is independent of temperature (see Supplementary Note 1). The vacuum-limited lifetime, as measured for $^{87}$Rb atoms, is $\gtrsim 100$ s; we therefore exclude this from our model and assume that a single process (with corresponding $\gamma$) dominates over the time scale of the measurement. We fix the initial temperature, and hence the initial density, in the fitting. An example result is shown in Fig. 1. We find an optimal value of $\gamma = 2.07(7)$ (reduced chi squared $\chi^2_{red} = 0.998$), shown by the solid black line, suggesting that the loss is governed by a two-body process. Fits with $\gamma$ fixed at 1, 2 and 3 have $\chi^2_{red} = 22.9$, 1.27 and 10.4, respectively; in all future fitting we therefore constrain the fits such that $\gamma = 2$. The results shown yield a two-body inelastic loss rate coefficient $k_2 = 4.8(6) \times 10^{-11}\ cm^3\ s^{-1}$.

To confirm the second-order behaviour, we explore the initial loss rate as a function of the starting density, by varying the number of molecules with the temperature and trap frequencies fixed. To extract the loss rate, we fit the first 0.2 s of molecule loss

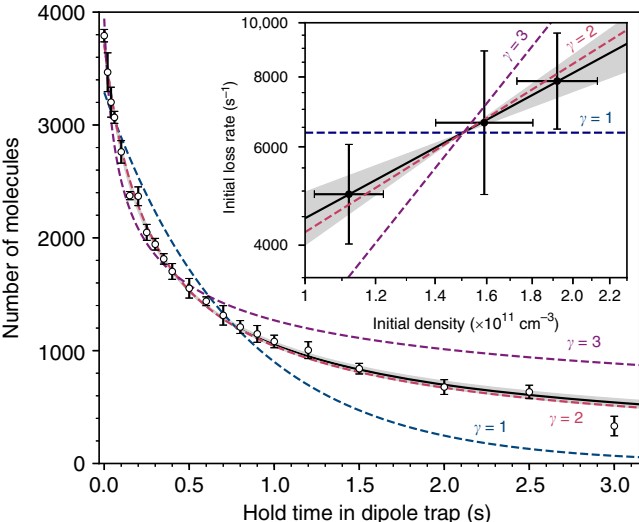

**Fig. 1** Loss of ground state molecules. Collisional loss of molecules in $|N = 0, M_N = 0, m_i^{Rb} = 3/2, m_i^{Cs} = 7/2\rangle$ with initial temperature of 1.5(1) µK and peak density $1.9(2) \times 10^{11}$ cm$^{-3}$. Each result is the mean of at least five experimental runs, with standard error shown. The solid black line shows a fit to the coupled rate equations given in Eq. (2) with 1 standard deviation ($\sigma$) uncertainty in $\gamma$ shaded. The dashed lines show fits to the data with fixed $\gamma = 1$, 2 and 3 corresponding to one-, two- and three-body loss, respectively. Inset: Density dependence of the initial loss rate on a log–log scale. The vertical error bars show the 1$\sigma$ uncertainty in the linear gradient fitted to the first 200 ms of each loss measurement, and the horizontal error bars show the 1$\sigma$ uncertainty in the density derived from the uncertainties in the starting temperature, number and trap frequencies. The solid line is a linear fit, with 1$\sigma$ uncertainty in the gradient shaded, while the dashed lines indicate the expectations for one-, two- and three-body loss

with a linear function to extract the gradient. The variation of the initial loss rate as a function of initial density is shown inset in Fig. 1 on a log–log scale. A linear fit to Fig. 1 yields a gradient of 0.9(3), again indicating a second-order process.

**Single-channel model and universal limit.** For a complex system such as RbCs + RbCs, it is not feasible to solve the many-dimensional Schrödinger equation directly using coupled-channel methods to extract rates for elastic and inelastic collisions. In this case, considerable success has been achieved with single-channel models based on quantum defect theory (QDT)[51,52]. These models take account of the fact that, at low-collision energy, much of the particle flux is reflected by the long-range attractive potential and never reaches short range. The probability of loss for particles that do reach short range is characterised by the parameter $y$, which is 1 when there is complete loss and 0 when there is no loss. Any resonant effects in the incoming channel are characterised by the short-range phase shift $\delta^s$, which in the absence of loss is related to the scattering length $a$ by $a/\bar{a} = 1 + \cot(\delta^s - \frac{\pi}{8})$. In the limit $y \to 1$, known as the universal limit, the loss is independent of $\delta^s$[51]. The universal rate coefficient at zero temperature is $k_2^{univ}(0) = 8h\bar{a}/m$ for identical bosons, where $\bar{a} = 0.477988\cdots \times (mC_6/\hbar^2)^{1/4}$ is the mean scattering length of Gribakin and Flambaum[53].

The long-range interactions are represented by their leading term $-C_6R^{-6}$, which is caused by the dispersion interaction. For RbCs in its rovibrational ground state, $C_6 = 141000\,E_h a_0^6$, which gives $\bar{a} = 233\,a_0$ and $k_2^{univ}(0) = 1.79 \times 10^{-10}$ cm$^3$ s$^{-1}$ at zero temperature. Our model[52] carries out QDT using Gao's analytic wavefunctions for a pure $R^{-6}$ potential[54,55], which account for

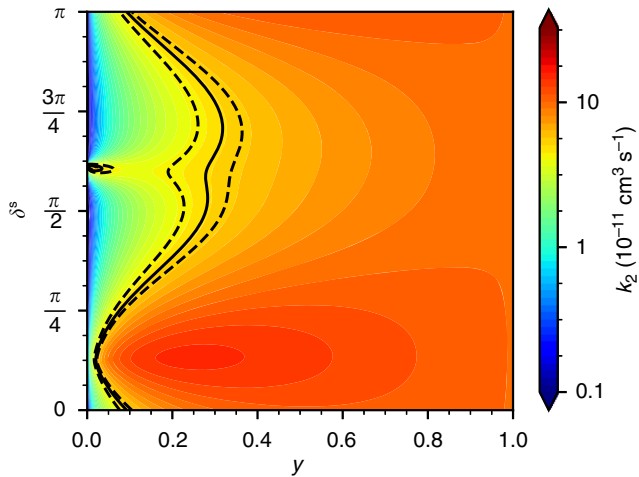

**Fig. 2** Thermally averaged loss rate coefficient from the single-channel model at 1.5 µK. $k_2(T)$ is plotted as a function of the loss parameter $y$ and the short-range phase shift $\delta^s$. The solid and dashed black lines correspond to the measured $k_2$ and uncertainty, respectively

reflection from the long-range potential. It allows variation of the loss parameter $y$ and includes multiple partial waves, so gives the complete energy dependence of the loss rates, rather than just the leading term as in ref. [51]. We calculate the thermally averaged rate coefficient, $k_2(T) = \int_0^\infty (2/\sqrt{\pi})k_2(E)x^{1/2}\exp(-x)\,dx$, where $x = E/k_B T$.

Figure 2 shows a contour plot of the thermally averaged loss rate coefficient $k_2(T)$ for ground-state RbCs + RbCs at 1.5 µK. Finite-temperature effects are important: In the universal limit, $y = 1$, the rate coefficient approaches $9.93 \times 10^{-11}$ cm$^3$ s$^{-1}$, which is nearly a factor of two lower than the zero-temperature value. When $y < 1$, the loss may be either lower or higher than the universal limit, depending on $\delta^s$. Around $\delta^s = \pi/8$, resonant s-wave scattering enhances the magnitude of the wavefunction at short range and causes a broad enhancement in the loss. Only even partial waves contribute for identical bosons. Around $\delta^s = 5\pi/8$ there is a narrower band of enhanced rates due to a d-wave shape resonance. Shape resonances for higher partial waves exist in $k_2(E)$[52], but are washed out by thermal averaging in $k_2(T)$.

Contours corresponding to the measured $k_2$ at 1.5 µK and its 1$\sigma$ confidence limits are shown in Fig. 2. There is a band of parameter space that gives loss rates in agreement with experiment. The largest part of this band is in the region $0.2 < y < 0.4$, but lower values of $y$ are possible in the region of large scattering length around $\delta^s = \pi/8$. Nevertheless, the region of agreement with the experiment is entirely $y < 0.4$, showing that this system is significantly removed from the universal limit ($y = 1$).

**Temperature dependence.** The temperature dependence of the loss rate contains important additional information. Figure 3 shows the calculated thermally averaged rate coefficients, as a function of temperature, for different values of the short-range phase $\delta^s$; for each phase, the loss parameter $y$ is chosen to match the experimental rate coefficient at $T = 1.5$ µK. It may be seen that the form (and even the sign) of the temperature dependence varies substantially with $\delta^s$.

We measure loss with a range of starting temperatures from 0.85(5) to 3.3(3) µK. Including temperature dependence allows us to fit the short-range phase as well as the loss parameter $y$. The best-fit parameters are $y = 0.26(3)$ and $\delta^s = 0.56^{+0.07}_{-0.05}\pi$

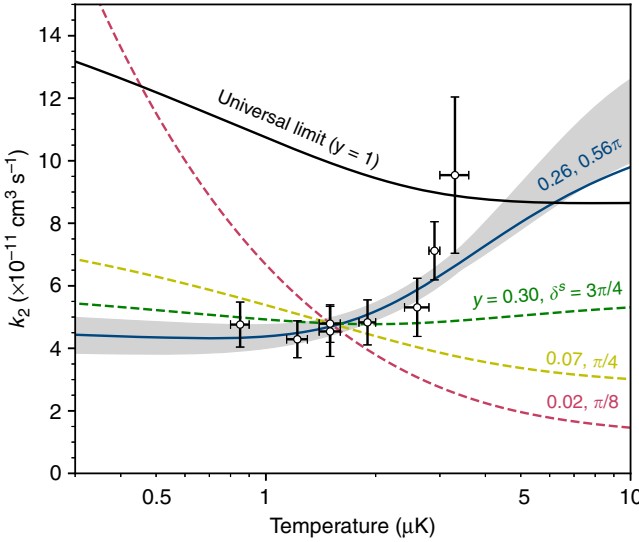

**Fig. 3** Temperature dependence of the loss rate coefficient. The horizontal and vertical error bars show the $1\sigma$ uncertainties in the measured temperature and loss rate coefficient $k_2$, respectively. The coloured dashed lines all match the experimental rate coefficient at $T = 1.5\,\mu$K, for a range of values labelled by $y$, $\delta^s$ that follow the solid black line in Fig. 2. The solid blue line shows the best fit, and the shaded region corresponds to the range of parameters that agree with the experimental results (open circles). The black line shows the loss rate in the universal limit ($y = 1$)

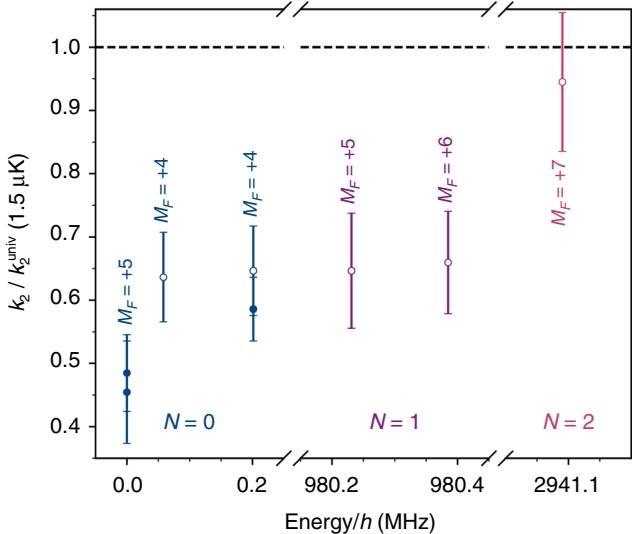

**Fig. 4** Loss rates as a fraction of the universal limit in a range of rotational and hyperfine states. Molecules are prepared in a state ($N$, $M_F$) with $T = 1.5$ (1) $\mu$K. The energy of each state is given relative to that of (0, +5). filled circles are measurements where the state is populated directly by STIRAP, whereas empty circles show where the molecules are transferred to the state with coherent microwave $\pi$-pulses (see methods). Error bars show the $1\sigma$ uncertainty in the measured rate constant $k_2$. Numerical values for the results are given in Supplementary Table 1

($\chi^2_{\text{red}} = 0.473$), and the fitted loss rate is shown as the blue line in Fig. 3 with uncertainty given by the shaded region. This gives us our first indication of the scattering length for RbCs + RbCs collisions; the fitted $\delta^s$ corresponds to $231\,a_0 < a < 319\,a_0$.

**Magnetic field dependence**. The single-channel model has no explicit dependence on magnetic field, but at fields below 98.8 G the initial state is no longer the lowest in energy. If hyperfine-changing collisions were a significant source of loss in the ground state, we would expect the loss rate to rise at lower fields. Conversely, if the loss is entirely mediated by the formation of collision complexes, it is unlikely to be affected by small changes in the energy of the asymptotic states and the loss rate will be independent of magnetic field.

We have measured loss of molecules at various magnetic fields between 4.6 and 229.8 G, and over this range the loss rate does not vary outside experimental uncertainties (see Supplementary Note 2). This suggests that loss due to hyperfine-changing collisions is not significant, and is consistent with the sticky collision hypothesis.

**Collisions in rotationally excited states**. We also consider loss of molecules in rotationally and hyperfine excited states[56]. We have measured loss rate coefficients at 1.5 $\mu$K for two hyperfine-excited states with $N = 0$, two states with $N = 1$ and one state with $N = 2$. The universal rate changes between states because of different rotational contributions to $C_6$, as shown in Supplementary Table 1. The rate coefficients as a fraction of the universal rate are shown in Fig. 4, labelled by $N$ and the total angular momentum projection $M_F = M_N + m_I^{\text{Rb}} + m_I^{\text{Cs}}$. This fraction is greater for excited states than for the ground state, and markedly greater for $N = 2$. The loss from the $N = 2$ state is consistent with the universal limit. This increase probably results from two-body inelastic collisions. However, the higher angular momentum also allows the incoming channel to couple to a larger number of

states of the collision complex[49], which might enhance the complex-mediated loss.

**Collisions in a mixture of rotational states**. We have also measured loss from an incoherent mixture of the spin-stretched states $N = 0$, $M_F = +5$ and $N = 1$, $M_F = +6$. These two states are linked by a dipole-allowed transition, so collisions between them experience an additional resonant dipole–dipole interaction. This is equivalent to the interaction of two space-fixed dipoles $d = d_0/\sqrt{6} = 0.50$ D. For s-wave scattering this interaction cancels in first order due to spherical averaging, but for higher partial waves it dies off asymptotically as $R^{-3}$. Even for s-waves, there are strong higher-order effects with leading term proportional to $R^{-4}$. These terms die off much more slowly than dispersion forces at long range, so may be expected to produce larger loss rates.

We start with a 50:50 mixture of molecules in the two states, at $T = 1.5(1)\,\mu$K, and measure the number remaining in each state as a function of time. We model the rates of change of the densities $n_0(\mathbf{r}, t)$ and $n_1(\mathbf{r}, t)$, for molecules in $N = 0$ and $N = 1$, respectively, by the coupled rate equations

$$\dot{n}_0(\mathbf{r}, t) = -k_2^{00} n_0(\mathbf{r}, t)^2 - \tfrac{1}{2} k_2^{01} n_0(\mathbf{r}, t) n_1(\mathbf{r}, t),$$
$$\dot{n}_1(\mathbf{r}, t) = -k_2^{11} n_1(\mathbf{r}, t)^2 - \tfrac{1}{2} k_2^{01} n_0(\mathbf{r}, t) n_1(\mathbf{r}, t). \tag{3}$$

We use the values of $k_2^{00}$ and $k_2^{11}$ measured above for molecules in identical rotational states. Fitting yields a value $k_2^{01} = 7.2(9) \times 10^{-10}$ cm$^3$ s$^{-1}$ for the loss rate coefficient for collisions between molecules in different rotational states. This is significantly higher than for molecules prepared in a single rotational and hyperfine state and demonstrates a significant increase in the loss rate due to a resonant dipole–dipole interaction.

**Discussion**

We have presented experimental measurements of loss rates for nonreactive RbCs molecules. We have demonstrated that the loss is best described by second-order rate equations. This suggests

that the loss is governed by a two-body process and supports the sticky collision hypothesis that the rate-limiting step is formation of long-lived collision complexes[48,49]. Through investigating the loss from the rotational and hyperfine ground state over a temperature range of 0.85(5) to 3.3(3) μK, we have determined the loss probability parameter $y$ at short range to be less than 0.4. We observe no change in loss rate with varying magnetic field. For rotationally excited states, the loss is up to a factor of 2 faster, probably due to rotational relaxation. For a mixture of rotational states, the loss is much faster, because of resonant dipole interactions.

Our results for the ground state are inconsistent with the universal limit of complete loss at short range ($y = 1$). The model of Mayle et al.[48,49] gives a loss rate that is independent of the density of states, provided the density is large and the average width is related to the average spacing as given by RRKM theory. In the ultracold limit, this rate is equivalent to the universal rate. The lower loss probability that we observe indicates that the average width is smaller than predicted by RRKM theory. This demonstrates a breakdown of ergodicity. A possible interpretation is that complex formation can occur only when the molecules collide at a limited range of relative orientations.

Our value of the loss parameter $y$ is similar to that seen for reactions of the type (1) in fermionic KRb ($y \sim 0.4$)[43,51], suggesting that a similar geometric restriction might apply in that case. Takekoshi et al.[3] published results with RbCs at a temperature of 8.7(7) μK, which is significantly higher than the present work. At fields above 90 G, they observed $k_2 \sim 1 \times 10^{-10}$ cm³ s⁻¹, which is consistent with both our fitted values of $y$ and $\delta^s$ and the universal limit. They also reported an increase in the loss rate by an order of magnitude at lower fields, which they attributed to hyperfine-changing collisions to form lower-energy states. However, as shown in the Supplementary Fig. 3, the increased loss rates are larger than the maximum allowed by reflection off the long-range potential at the temperature of the experiment. The only other nonreactive molecule for which collisions have been studied in detail is NaRb; the results were interpreted as consistent with the universal limit[47,57], but the observed temperature dependence resembles that calculated here for resonant s-wave scattering at lower $y$, as shown by the orange line in Fig. 3.

In conclusion, our measurements of collisional losses in ultracold RbCs support the sticky collisions hypothesis, but the rates are significantly lower than the universal limit. By examining the temperature dependence, we have seen the first indication of the scattering length for RbCs + RbCs collisions. By preparing an incoherent mixture of ground and first-excited rotational states, we turn on a resonant dipole–dipole interaction which greatly increases the loss rate. Our results indicate that active measures to suppress collisional loss will be needed in experiments with high-density molecular gases, even if the molecules are nonreactive.

## Methods

**Transfer of molecules to the ground state.** We begin our experiments with a sample of weakly bound RbCs Feshbach molecules[58], confined to a $\lambda = 1550$ nm ODT at a magnetic field of 181.5 G. We transfer the molecules to a single hyperfine level of the $X^1\Sigma^+$ rovibrational ground state via stimulated Raman adiabatic passage (STIRAP)[4] in free space (i.e., with the trap light off) to avoid a spatially varying ac Stark shift of the two-photon resonance[59]. The efficiency of the STIRAP is typically 90%, and we can transfer to hyperfine states in $N = 0$ with $M_F = +5$ or $M_F = +4$ depending on the selected laser polarisation[60]. Following STIRAP, the molecules are recaptured by turning the trapping light back on. We set the intensity of the trap light before ($I_{FB}$) and after ($I_{GS}$) STIRAP such that the ground-state molecules experience the same trap parameters as they did in the Feshbach state, i.e., $I_{GS}/I_{FB} = \alpha_{FB}/\alpha_{GS}$ where $\alpha_{FB}$, $\alpha_{GS}$ are the polarizabilities of the Feshbach and ground states[61]. The ground-state transfer takes 20 μs and the trap light is off for less than 200 μs in all experiments presented; we detect no significant heating or loss from this modulation of the trap potential.

**Detection of molecules.** We measure the number of molecules by reversing the association process, dissociating the molecules back to their constituent atoms which are detected via absorption imaging. We, therefore, only image molecules which occupy the specific hyperfine state accessed through the STIRAP. We extract the number from each absorption image either by summing pixels in a fixed region of interest or by least squares fitting to a 2D Gaussian function. We find similar numbers using both methods. Results plotted in this work show the numbers found using the pixel summing algorithm.

We produce up to $N_{mol} = 4000$ ground-state molecules. By varying the hold time between the ground-state molecule recapture and the dissociation for imaging, we record the time evolution of the number of molecules remaining in the dipole trap.

**Measurement of trap frequencies.** We measure the trap frequencies experienced by the molecules by observing centre-of-mass oscillations in the optical potential. We also compare the oscillation frequencies for the molecules to those of atoms in the same potential[61]. For the results shown in Fig. 1, we find $(\omega_x, \omega_y, \omega_z) = 2\pi \times (181(2), 44(1), 178(1))$ Hz, where $z$ is in the direction of gravity.

**Measurement of temperatures.** The initial temperature of the molecules is measured by ballistic expansion in free space. Due to the small number of molecules, we can only image the cloud over an expansion time of ~2 ms. For comparison, we also measure the temperature of atoms by the same method in similar trapping conditions. We find good agreement between the temperature of the molecules and that of the atoms. We do not measure the variation of temperature as a function of time during the loss measurement, as the loss of molecules further limits the maximum expansion time available leading to unreliable temperature measurements.

The rate Eq. (2), which we use to model the loss, depends on both $N_{mol}(t)$ and $T$ ($t$). As described in the main text, we fit $N_{mol}(t)$ for a fixed initial $T$, allowing the temperature to evolve as a function of time within the constraints of the model. We have also fitted our results assuming the molecules remain at their initial temperature throughout the measurement. In this limit, we find that our results are still consistent with a two-body process. For the results in Fig. 1, we extract $k_2 = 3.8$ (5) $\times 10^{-11}$ cm³ s⁻¹.

**Optical trapping and varying temperature.** To vary the temperature, we adiabatically compress the molecules prior to ground-state transfer. The lowest temperature measurements we perform use the $\lambda = 1550$ nm ODT in which the Feshbach molecules are initially prepared. The trap light is derived from a single-mode IPG fibre laser, which is split into two beams with focused waists 80 and 98 μm crossing at an angle of 27.5°. There is a frequency difference of 100 MHz between the two beams originating from the acousto-optic modulators used to control the beam intensities independently. In this trap we can access temperatures from 0.85(5) to 1.9(1) μK for geometrically averaged trap frequencies $\bar{\omega}/(2\pi)$ between 79 and 149 Hz. This trap is used for all loss measurements with a temperature of 1.9 μK or below.

To explore higher temperatures, we transfer the molecules to a different optical potential with $\lambda = 1064.52$ nm. The light is generated by a Coherent Mephisto master oscillator power amplifier, and the trap formed by crossing two beams with focused waists 64 and 67 μm (and 160 MHz frequency difference) at an angle of 54°. To transfer the molecules between the two traps, we ramp the powers linearly over 50 ms. In this trap, we performed measurements at temperatures of 2.6(2) and 3.3(3) μK. The result in Fig. 3 with $T = 2.9(1)$ μK is performed with a mixed wavelength potential, using one beam of the $\lambda = 1064$ nm trap crossed with one beam of the $\lambda = 1550$ nm trap. This removes the possibility of loss or heating due to the 100/160 MHz beat frequency between the two beams, and allows us to rule out intensity-dependent losses from either trap.

**Eliminating other sources of loss.** Collisions of ground-state molecules with Rb atoms, Cs atoms or molecules in excited states could also cause loss. Following Feshbach association, we remove the remaining Rb and Cs atoms from the trap via the Stern–Gerlach effect. During the separation the atoms do not experience a trap for over 20 ms, which is sufficient to ensure that all atoms have left the region of interest. The STIRAP process is typically 90% efficient, with the 'lost' molecules likely being addressed by the pump light and transferred to the $^3\Pi_1$, $v = 29$, $N = 1$ electronically excited state. The lifetime for molecules in this excited state is 16(1) μs[59], following which the molecules may decay to either $a^3\Sigma^+$ or $X^1\Sigma^+$. We have performed measurements with STIRAP efficiency between 79% and 93% with no measurable change to the loss rate indicating that the molecule fraction which is not transferred to the ground state plays no role in the subsequent loss. This is consistent with similar observations in NaRb[47].

We have observed narrow resonant loss features around 1064.48 nm which are dependent on the laser frequency and intensity. We have investigated the intensity and density dependence of these features and conclude that they result from two-photon excitation of the molecules. All measurements using the 1064 nm trap are performed at a wavelength of 1064.52 nm, sufficiently far from the narrow loss features to remove them as a source of loss. No additional loss features were observed when trapping with 1550 and 1064 nm light together.

We also discount other light-scattering losses in our experiments. Loss of molecules due to the absorption of black-body radiation has a rate of $10^{-5}\,\mathrm{s}^{-1}$ for RbCs at room temperature[62]. For laser light with $\lambda = 1550$ nm, the photon energy is greater than the dissociation energy of the electronic ground state but far below the potential minimum for the $b^3\Pi$ state. Photons of wavelength 1064.52 nm are above the potential minimum of the $b^3\Pi$ state, but transitions to the accessible vibrational levels are strongly suppressed due to small Franck–Condon factors[63]. By performing measurements at these two trapping wavelengths, we demonstrate that the loss we observe is independent of the wavelength of the trap light. Moreover, by using a mixed-wavelength trap, we eliminate the possibility of intensity dependent losses.

**Internal state control and transfer**. Following the ground-state transfer, we pulse on microwaves to perform either a single or a pair of coherent $\pi$-pulses, transferring the molecules to a different rotational and/or hyperfine state. The microwave transfer is performed with the optical trap off, and has unit efficiency[56]. We tune the intensity of the microwaves such that the Rabi frequency is small enough to avoid off-resonant excitation of nearby transitions, while still obtaining a $\pi$ pulse duration of <100 µs. To read out the number of molecules in an excited state we must reverse the sequence of $\pi$-pulses to transfer back to the original state used for STIRAP.

Measuring loss in higher rotational states requires a good understanding of the molecular polarizability, and hence the trapping potential observed for each state. The trapping light is linearly polarised parallel to the magnetic field, and in this case the states chosen each have a linear ac Stark shift as a function of laser intensity. This is necessary to avoid possible Landau–Zener-type loss associated with avoided crossings between hyperfine states[61]. For each state, we tune the intensity of the optical trap so that the molecules always experience the same trap frequency and depth as they do in the ground state. We do not expect any spontaneous emission from the rotationally excited states as the rate is ~$10^{-5}\,\mathrm{s}^{-1}$.

**Preparation of an incoherent mixture**. To generate a 50:50 mixture of molecules in different rotational states, we drive a $\pi/2$-pulse on the transition between $N=0$, $M_F = +5$ and $N=1$, $M_F = +6$ in free-space. This puts the molecules into a coherent, equal superposition of the two states. The molecules are recaptured in the $\lambda = 1550$ nm ODT, where the superposition rapidly dephases due to spatial variation in the energy difference between the states[64]. Using Ramsey spectroscopy, we observe no signs of coherence after a 10 µs hold in the ODT; this is four orders of magnitude faster than the timescale of the loss. The density matrix which describes the cloud following this dephasing contains only the diagonal elements and thus can be considered a mixed state.

As the two states have different polarizabilities, $\alpha_{N=1}/\alpha_{N=0} \approx 0.9$, we cannot tune the laser intensity to match the trap parameters to those before preparation for both states. We have performed experiments where the trap frequency and depth is matched for either $N=0$ and $N=1$, and we measure the same value of $k_2$ in both cases.

**Dispersion coefficients**. Dispersion coefficients $C_6$ arise from the dipole-dipole interaction in second order and may be calculated using perturbation theory. For the interaction of two RbCs molecules, they are dominated by rotational terms involving the permanent molecular dipole moment. The necessary matrix elements can be found in, for example, ref. [65]. The result for the rotational ground state, $C_{6,\mathrm{rot}} = \mu_{\mathrm{elec}}^4/6B$, is well-known. For rotationally excited states, the diagonal part of $C_6$ varies with the projection quantum number $M_N$ and with partial wave $L$. There are additional contributions to $C_6$ from electronic dispersion and induction interactions, which we take from ref. [66].

We calculate the $C_6$ coefficients using accurate values for the RbCs electric dipole moment $\mu_{\mathrm{elec}} = 1.225$ D[4] and rotational constant $B = 490.173994$ MHz[56]. For the rotational ground state the combination of rotational and electronic contributions gives $C_6 = 141000\,E_\mathrm{h}a_0^6$. For the rotationally excited state, we find for $L = 0$: $C_6 = 141000\,E_\mathrm{h}a_0^6$ for $N = 1$, $M_N = 0$; $C_6 = 96000\,E_\mathrm{h}a_0^6$ for $N = 1$, $M_N = 1$; and $C_6 = 82000\,E_\mathrm{h}a_0^6$ for $N = 2$, $M_N = 2$.

## Acknowledgements

This work was supported by the U.K. Engineering and Physical Sciences Research Council (EPSRC) Grants nos. EP/H003363/1, EP/I012044/1, EP/P008275/1 and EP/P01058X/1. The authors thank A. Kumar, P. K. Molony and Z. Ji for their work in the early stages of the experiment, J. Aldegunde for calculations regarding the hyperfine and rotational structure of RbCs and T. Karman for valuable discussions.

## Author contributions

P.D.G., J.A.B., E.M.B. and R.S. performed the experiments, and are responsible for analysis in determining second-order behaviour and extracting loss rates. M.D.F. carried out the single-channel calculations. S.L.C. and J.M.H. supervised the project. The paper was written by P.D.G., M.D.F., J.M.H. and S.L.C.

## Additional information

**Competing interests:** The authors declare no competing interests.

**Peer review information**: Nature Communications thanks the anonymous peer reviewers for their contributions to the peer review for this work. Peer review reports are available.

## Data availability

All data presented in this work are available at https://doi.org/10.15128/r270795768f.

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
