## [Peer Review File · Nature Communications]

Reviewers' comments:

Reviewer #1 (Remarks to the Author):

In this manuscript, the authors presented a careful study on the collisional loss of ultracold ground-state RbCs molecules. Ultracold polar molecules are of great current interest due to its promising application in quantum science and quantum physics. Currently, there are several issues which are greatly slowing down the progress. The first is the difficulty in producing the ultracold sample of ground-state polar molecules. The second is the problem of loss which persists regardless of the chemical reactivity. Unfortunately, due to the first reason, the investigation on the loss is far from complete.

The Durham group is one of the handful groups which now have the capability of producing ultracold sample of ground-state polar molecules. The work presented in the paper is much needed for understanding of the loss problem without known chemical reaction channels. The data shown in the paper are all very clear, and the data analysis seems convincing. Especially, the comparison between different internal state combinations should be very valuable for many other groups to understand the issue.

The manuscript is well organized and presented in a very clear way. I thus recommend it to be accepted as is.

Reviewer #2 (Remarks to the Author):

In the manuscript, the authors present observations that support the hypothesis of long-lived complexes in collisions of ultracold molecules. These complexes have been suggested to lead to "sticky" collisions, which could result in strong losses of ultracold molecules even when they are non-reactive. The authors further find that the loss rate is slower compared to the limit of 'universal loss.' By comparison to predictions from a single-channel quantum defect theory (QDT) model, the authors extract quantitative parameters (loss parameter and scattering phase shift) of 87Rb-133Cs collisions in this framework.

Whether "sticky collisions" occur in non-reactive bi-alkalis have been a long-standing question for a good part of the past decade. A clear answer to this question is of increasing urgency, as ultracold molecular gases are poised to become a new quantum resource for quantum simulation in the near future. The work that is presented here is an important step in establishing the presence of these collisions. It goes beyond previous work by quantitatively estimating a loss probability in a simple effective QDT model. I find the manuscript to be well-written and the experiment to be analyzed carefully and thoroughly. I fully support publication of the manuscript in nature communications with minor modifications listed below. The authors should also address concerns listed below.

Concerns/Questions:

- 1) The single-body lifetime (vacuum-limited ideally) should be stated to justify fitting to loss rate with a single value of γ . I assume it is much longer than 3 seconds, since it is not included in the fit?
- 2) The analysis assumes that the loss coefficient is independent of temperature. Yet in the comparison to theory, a thermal average is taken, and the authors state that "finite temperature effects" are important. So how much does the the temperature vary according to the fitted parameters and model assuming fixed γ ? The authors should comment on this, and provide a curve of the evolution of temperature (according to the fitted model) in the Supp. Mat.
- 3) Does one have to worry about two-photon transitions to a different vibrational state when both 1064 and 1550nm trapping beams both on, or is the energy difference sufficiently large or are the Franck-Condon factors sufficiently small?

Minor suggested modifications:

- 1) In the section "measuring loss due to molecule-molecule collisions," the authors state the assumption that the molecules remain in thermal equilibrium. Maybe the authors can support this assumption by a) estimating that the number of elastic collisions is sufficient to establish local equilibrium and b) that the timescale of evolution of molecular number is much larger than a quarter of the trapping period?
- 2) In the penultimate paragraph in the section "Single-channel model" and universal limit, it might be useful to remind the reader that only s and d-wave scattering is allowed since the molecules are bosonic.
- 2) In the same section, the last sentence in the last paragraph, one could state " $\gamma=1$ " after the universal limit to help the reader.
- 3) In the section "Temperature dependence," the phrase "indication of the scattering length" is unclear. Maybe the authors mean the first experimental indication of the value of the scattering length? (same comment applies to the last paragraph in the section "discussion.")
- 4) In the same paragraph: $231 100 s; we therefore exclude this from our model and assume that a single process (with corresponding γ) dominates over the time scale of the measurement."

- 2) **The analysis assumes that the loss coefficient is independent of temperature. Yet in the comparison to theory, a thermal average is taken, and the authors state that "finite temperature effects" are important. So how much does the temperature vary according to the fitted parameters and model assuming fixed γ ? The authors should comment on this, and provide a curve of the evolution of temperature (according to the fitted model) in the Supp. Mat.**

As the reviewer has suggested, we have added Supplementary Fig. 1, which shows the time evolution of the temperature according to the fitted model for each experimental measurement in Fig. 3. The variation in temperature over a single measurement is typically ~ 1 μK . As we have no method of experimentally verifying this temperature change, we use the starting temperature in our analysis. For our best-fit parameters, shown in Fig. 3, the loss rate is relatively flat as a function of temperature, certainly below 2.5 μK , and we therefore do not expect this assumption to have much impact. We have added a paragraph discussing this to supplementary note 1.

- 3) **Does one have to worry about two-photon transitions to a different vibrational state when both 1064 and 1550nm trapping beams both on, or is the energy difference sufficiently large or are the Franck-Condon factors sufficiently small?**

We expect that the linewidth for transitions between vibrational states would be very narrow. When preparing to measure the lifetime in the 1064nm + 1550nm crossed trap, we varied the frequency of the 1064 laser over several GHz. This allowed us to eliminate the possibility of loss due to the driving of a vibrational transition. We have added a statement in the Methods to clarify this.

Minor suggested modifications from reviewer #2:

- 1) **In the section "measuring loss due to molecule-molecule collisions," the authors state the assumption that the molecules remain in thermal equilibrium. Maybe the authors can support this assumption by a) estimating that the number of elastic collisions is sufficient to establish local equilibrium and b) that the timescale of evolution of molecular number is much larger than a quarter of the trapping period?**

Using the single-channel QDT model, we can calculate elastic cross sections as well as loss rates. For our best-fit parameters, the elastic cross section is larger than the loss cross section by a factor of 1.8 at $E = 1.5 \mu\text{K} \times k_B$. In addition, the timescale of the loss is significantly greater than a quarter of the trapping period for all measurements performed. We have added a paragraph addressing these points to supplementary note 1.

- 2) **In the penultimate paragraph in the section "Single-channel model" and universal limit, it might be useful to remind the reader that only s and d-wave scattering is allowed since the molecules are bosonic.**

We have added a reminder that only even partial waves are allowed for identical bosons.

- 3) **In the same section, the last sentence in the last paragraph, one could state " $\gamma=1$ " after the universal limit to help the reader.**

We have added this.

- 4) **In the section "Temperature dependence," the phrase "indication of the scattering length" is unclear. Maybe the authors mean the first experimental indication of the value of the scattering length? (same comment applies to the last paragraph in the section "discussion.")**

This is indeed the first indication of the value of the scattering length, from either experiment or theory. The scattering length is unobtainable from pure theory, because no interaction potential would correctly predict the position of the least-bound state. We do not wish to say that this is the first "experimental indication", because we obtain it from a combination of experiment and theory. We therefore prefer to keep "first indication of the value of the scattering length".

- 5) **In the same paragraph: $231 \ 230 \ a_0 < a < 319 \ a_0$.**

We have corrected this.

- 6) **Figure 3: I suggest reverse ordering of the 3 sentences in the caption to improve readability. Also, in the figure itself, it helps to have γ and δ 's written for at least one of the dotted curves. E.g. $\gamma=0.26$, $\delta=0.5\pi$.**

We have reordered the caption for better readability as suggested, and γ and δ 's labels have been added for the green dashed line.

- 7) **In the supplementary materials, the description of the E_{col} is not clear. The authors should state that E_{col} is the total energy of colliding molecules averaged over all molecules. It is easy to misread this as the collision energy.**

We have changed the notation to E_{avg} to avoid confusion, and better explained the quantity when it is introduced in supplementary note 1.

- 8) **The result of E_{col} in Eq. (3) is correct, but the middle part is not. The middle part only contains the potential energy, and is missing the kinetic part. This should be corrected. It also does not hurt to mention the two contributions. (supplementary material)**

We have corrected this and the potential and kinetic contributions are now stated explicitly.

- 9) **In Eq. (4), the result is correct, but the middle part is missing a minus sign. The authors could also point out that for $\gamma=1$, which corresponds to one-body loss, the temperature is unchanged. This follows since, in this case, both the loss rate and temperature are intensive, and does not depend on the total number of particles. (supplementary material)**

We have corrected this and have added that for one-body loss the temperature is unchanged.

Reviewer #3 response:

This paper addresses one of the important questions in cold molecule research: what causes loss of trapped molecules even in the absence of any two-body inelastic collisions? Understanding this loss process is important for many applications of cold molecules. An often-invoked hypothesis is “sticky collisions” which is another terminology for complex formation. The high density of states of heavy dimers such as RbCs can form pairs of collision complexes (Feshbach resonances) which upon collisions with another molecule may lead to the loss of all three molecules from the trap. This “sticky collision” hypothesis is verified by combined experimental and theoretical studies of ultracold trapped RbCs molecules offering significant new insights into lifetime of trapped molecules. It is found that the loss process can be described by a rate equation that is second-order in molecule density but the observed rate is lower than that predicted by the universal model. The paper is well-written, highly relevant and impactful. I recommend it for publication in Nature Communications once the following comments are adequately addressed.

1. It will be useful to compare the measured lifetime against RRKM predictions. The RRKM predictions give a lifetime of 45 ms. With a measured loss rate coefficient of about $5 \cdot 10^{-11} \text{ cm}^3 / \text{s}$ and an initial density of $1.9 \cdot 10^{11} \text{ cm}^{-3}$ yields a lifetime of about 100 ms, roughly a factor of 2 longer than the RRK prediction. It is useful to highlight this as it provides direct experimental validation of lifetime estimates based on simple RRKM theory.

2. Ideally one would like to verify the complex formation hypothesis through explicit computation of Feshbach resonances. However, as correctly pointed out in this manuscript, it is not practical for heavy dimers such as RbCs. In this context, I would like to draw attention to a recent paper of Croft et al. (Phys. Rev. A 96, 062707 (2017)) which directly addressed this issue through explicit quantum scattering calculations of Feshbach resonances but for an atom-diatom system (K₂+Rb) which is computationally tractable. For the ground ro-vibrational level of K₂ only elastic scattering is possible but numerous Feshbach resonances are observed in this system due to coupling with closed channels of both K₂ and KRb. In this context, I would like to see authors' comments on whether energetically closed channels of Cs₂ or Rb₂ may involve in complex formation and whether inclusion of these molecular states in the density of states calculation may improve the agreement with experiment. In other words, though inelastic loss through chemical reaction is not energetically allowed coupling to reactive channels may still contribute to complex formation. Minor point: The expression of thermally averaged rate coefficient, $k_2(T)$ given in page 3 (right column, line #4 from top) appears to be missing a $k_B T$ in the denominator. Please check. Also, there is a typo in the units for the peak density, $1.9(2) \cdot 10^{11} \text{ cm}^{-3}$ (cm^3 instead of cm^{-3}) in the left column of page 2

Our responses to the reviewer's specific questions can be found below:

- 1) It will be useful to compare the measured lifetime against RRKM predictions. The RRKM predictions give a lifetime of 45 ms. With a measured loss rate coefficient of about $5 \cdot 10^{-11} \text{ cm}^3 / \text{s}$ and an initial density of $1.9 \cdot 10^{11} \text{ cm}^{-3}$ yields a lifetime of about 100 ms, roughly a factor of 2 longer than the RRK prediction. It is useful to highlight this as it provides direct experimental validation of lifetime estimates based on simple RRKM theory.

RRKM theory deals with the rate of decay (or lifetime) of a complex that has already been formed. The lifetime depends on the density of states, and within the model of Mayle et al. is 45 ms for $(\text{RbCs})_2$. However, the bimolecular decay rate in Mayle's model is equivalent to the universal rate and is independent of the density of states, provided the density is large. It comes principally from the rate of formation of complexes, rather than their decay. Our experiments indicate that the lifetime is longer than RRKM would predict for a given density of states, but they do not numerically determine the density of states or the lifetime. We have changed the second paragraph of the discussion to explain this point.

- 2) **Ideally one would like to verify the complex formation hypothesis through explicit computation of Feshbach resonances. However, as correctly pointed out in this manuscript, it is not practical for heavy dimers such as RbCs. In this context, I would like to draw attention to a recent paper of Croft et al. (Phys. Rev. A 96, 062707 (2017)) which directly addressed this issue through explicit quantum scattering calculations of Feshbach resonances but for an atom-diatom system (K_2+Rb) which is computationally tractable. For the ground ro-vibrational level of K_2 only elastic scattering is possible but numerous Feshbach resonances are observed in this system due to coupling with closed channels of both K_2 and KRb . In this context, I would like to see authors' comments on whether energetically closed channels of Cs_2 or Rb_2 may involve in complex formation and whether inclusion of these molecular states in the density of states calculation may improve the agreement with experiment. In other words, though inelastic loss through chemical reaction is not energetically allowed coupling to reactive channels may still contribute to complex formation.**

The answer to this is closely related to the previous one: any large density of states gives the same rate constant if the lifetime of the states is related to the density as given by RRKM theory. Our measurements throw light on whether the decay of the states is statistically related to the density, rather than quantifying the density. We thus cannot extract any information on the specific nature of the states, such as (but not limited to) the extent to which they reside in the "reactant" or "product" parts of the phase space. Such matters are also quite model-dependent, so we prefer not to comment on them in this paper.

Minor point:

The expression of thermally averaged rate coefficient, $k_2(T)$ given in page 3 (right column, line #4 from top) appears to be missing a $k_B T$ in the denominator. Please check. Also, there is a typo in the units for the peak density, $1.9(2) \cdot 10^{11} \text{ cm}^{-3}$ (cm^3 instead of cm^{-3}) in the left column of page 2

We have corrected the equation for $k_2(T)$ and rewritten it in terms of $x=E/(k_B T)$. Our calculations used the correct equation and are not affected. We have corrected the typo in the units for the peak density.

REVIEWERS' COMMENTS:

Reviewer #2 (Remarks to the Author):

The authors have adequately addressed all of my concerns. I recommend publication as is.

Reviewer #3 (Remarks to the Author):

This reviewer provided confidential remarks recommending publication.